# Sharding-Based Proof-of-Stake Blockchain Protocols: Key Components & Probabilistic Security Analysis [note 1]

**DOI:** 10.3390/s23052819

**Published:** 2023-03-04

**Authors:** Abdelatif Hafid, Abdelhakim Senhaji Hafid, Dimitrios Makrakis

**Affiliations:** 1Department of Computer Science and Operations Research, Université de Montréal, Montreal, QC H3T 1J4, Canada; 2School of Electrical Engineering and Computer Science, University of Ottawa, Ottawa, ON K1N 6N5, Canada

**Keywords:** security analysis, blockchain, probabilistic analysis, sharding-based blockchain protocols, malicious nodes, proof of stake, practical Byzantine fault tolerance

## Abstract

Blockchain technology has been gaining great interest from a variety of sectors including healthcare, supply chain, and cryptocurrencies. However, Blockchain suffers from a limited ability to scale (i.e., low throughput and high latency). Several solutions have been proposed to tackle this. In particular, sharding has proved to be one of the most promising solutions to Blockchain’s scalability issue. Sharding can be divided into two major categories: (1) Sharding-based Proof-of-Work (PoW) Blockchain protocols, and (2) Sharding-based Proof-of-Stake (PoS) Blockchain protocols. The two categories achieve good performances (i.e., good throughput with a reasonable latency), but raise security issues. This article focuses on the second category. In this paper, we start by introducing the key components of sharding-based PoS Blockchain protocols. We then briefly introduce two consensus mechanisms, namely PoS and practical Byzantine Fault Tolerance (pBFT), and discuss their use and limitations in the context of sharding-based Blockchain protocols. Next, we provide a probabilistic model to analyze the security of these protocols. More specifically, we compute the probability of committing a faulty block and measure the security by computing the number of years to fail. We achieve a number of years to fail of approximately 4000 in a network of 4000 nodes, 10 shards, and a shard resiliency of 33%.

## 1. Introduction

With the rise of Bitcoin [1], Blockchain has attracted significant attention from both industry and academia. More specifically, it has been adopted across different industries including healthcare [2,3], finance [4], and the public sector [5]. However, Blockchain suffers from poor scalability [6]. For example, in the case of cryptocurrencies, Bitcoin [1] handles between three and seven transactions per second (tx/s), which is very limited compared to traditional electronic payment systems (e.g., PayPal [7]). Several solutions were proposed to scale Blockchain. In particular, sharding has emerged as a promising solution [6]. Sharding consists of partitioning the network into sub-networks, called shards. All shards work in parallel to enhance the performance of the network. More specifically, each shard processes a sub-set of transactions instead of the entire network processing all the transactions. While sharding considerably improves scalability, it decreases the level of Blockchain security. More specifically, in sharding-based Blockchains, it is easier for a malicious user to attack and conquer a single shard compared to the whole network. This attack is well-known as a shard takeover attack (also referred to as a 1% attack) [8].

Blockchain networks are susceptible to sybil attacks by malicious nodes (called sybil nodes). Several consensus mechanisms (e.g., PoW, PoS, and pBFT) have been proposed to defend against these sybil nodes. Sharding-based Blockchain protocols [6,9] can be classified into two classes: sharding-based PoW and sharding-based PoS Blockchain protocols.

Recently, Hafid et al. [8,10,11] proposed mathematical models to analyze the security of sharding-based PoW protocols. However, to the best of our knowledge, there is no existing work that proposes a probabilistic security analysis of sharding-based PoS Blockchain protocols except an earlier version of this paper, which has been published in [12].

In this paper, we focus on the second category. We start by presenting the key components of these protocols. This article briefly describes PoS and pBFT consensus mechanisms, and discusses their use and limitations in the context of sharding-based Blockchain protocols. Finally, we propose a probabilistic model to analyze the security of these protocols by computing the probability of committing a faulty block. Based on these probabilities, we calculate the number of years to fail for the purpose of quantifying and measuring the security of the network.

The remainder of the paper is organized as follows. Section 2 presents the key components of sharding-based PoS Blockchain protocols and compares these protocols with related Blockchain protocols. Section 3 presents the proposed probabilistic model. Section 4 presents the numerical results and evaluates the proposed model. Section 5 present limitations of the paper and future work. Lastly, concluding remarks are given in Section 6.

## 2. Key Components of Sharding-Based PoS Blockchain Protocols

In this section, we cover the main components of shading-based PoS Blockchain protocols and compare these protocols with Bitcoin and sharding-based PoW Blockchain protocols.

### 2.1. Key Components

In this section, we shed light on the key components of sharding-based PoS Blockchain protocols. More specifically, we focus on Incognito [13] as an example. However, most of the sharding-based PoS Blockchain protocols share the same structure (see Figure 1). The key components of these protocols are:**Validator:** A node (can be malicious or honest) that competes to produce and add blocks to the blockchain.**Consensus mechanism:** Most of the sharding-based PoS Blockchain protocols use pPFT alongside PoS. PoS is an alternative consensus mechanism to PoW. Unlike PoW, which relies on miners with hash power to solve a mathematical puzzle, PoS relies on validators staking coins. Validators are selected based on the number of coins they stake (e.g., a validator with 10% of the total coins staked, has a 10% probability of adding a block to the blockchain). PoS consists of selecting validators in proportion to their number of coins. Validators are responsible for adding new blocks to Blockchain. pBFT is an algorithm that tolerates Byzantine faults [14]; An algorithm belonging to the Byzantine Fault Tolerance (BFT) class. BFT is the ability of the network to reach a consensus, on some value, even in the case of having some failed and/or malicious nodes in the network. Lamport et al. [15] proved that if we have 3m+1 correctly working processors, a consensus (agreement on same value) can be reached if at most *m* processors are faulty; this means that strictly more than two-thirds of the total number of processors should be honest.**Beacon Chain:** The main chain in the network. It is responsible for randomly assigning the validators to the committees of shards. The Beacon chain confirms cross-shard information and shuffles the committees of shards to ensure security.**Chain (aka shard):** A chain that consists of a subset of nodes of the network. Shards process different transactions in parallel to improve scalability. Generally, a sharding-based PoS network consists of one beacon chain and several shards.**Beacon Committee:** A subset of validators, selected randomly from the set of validators, that has decided to stake for the beacon chain. Generally, its main role is to check/verify and insert the valid block header into the beacon chain.**Shard Committee:** A subset of validators, selected randomly from the set of validators, that has decided to stake for the shard chain. A shard committee validates and confirms the transactions processed in the shard.

### 2.2. Comparison with Related Blockchain Protocols

To demonstrate the advantages and limitations of sharding-based PoS Blockchain protocols, Table 1 shows a comparison of its performance with sharding-based PoW Blockchain protocols and PoS Blockchain protocols. This comparison reflects that sharding-based PoS Blockchain protocols have several good features. Specifically, sharding-based PoS Blockchain protocols have higher transaction throughput, requires less computational power, and in some cases offers higher level of privacy (e.g., Incognito [13]) and lower transactions fees. However, some limitations still remain in sharding-based PoS Blockchain protocols; for example, centralization concerns caused by coinage.

Table 1 shows that by using sharding alongside PoS and pBFT consensus mechanisms, we get better performance without the need for high hash computing power. For example, the throughput of Incognito [13] scales out linearly with the number of shards. Specifically, Incognito achieves a significantly higher number of transactions (up to 800 tx/s for only 64 shards; thus, for 100 s of shards, the throughput will be in 1000 s tx/s). Zilliqa [16] handles up to 2800 tx/s [17].

**Table 1 sensors-23-02819-t001:** Bitcoin vs. Sharding-Based PoW Blockchain Protocols vs. Sharding-Based PoS Blockchain Protocols.

	Bitcoin [1]	Zilliqa [16]	Elastico [18]	Incognito [13]	Harmony [17]	Nxt [19]
PoW	✓	✓	✓	✕	✕	✕
PoS	✕	✕	✕	✓	✓	✓ ^1^
pBFT	✕	✓	✓	✓	✓	✕
Sharding	✕	✓	✓	✓	✓	✕
Throughput	Up to 7 tx/s	2800 tx/s	40 tx/s	100 ^1^ tx/s and 800 ^2^ tx/s	N/A	4 tx/s
Latency	∼1 h	N/A	800 s	N/A	N/A	∼10 min
Resiliency	Supports up to 50% (≰51%) of Byzantine fault	Up to 33% for shard’s committee and 25% for the entire network	Up to 33% for shard’s committee and 25% for the entire network	Up to 33% for shard’s committee and 51% for beacon’s committee	Up to 33% for shard’s committee and 25% for the entire network	Supports up to 33% (≰$\frac{1}{3}$) of Byzantine fault participants
Unique Features	Mining competition	Proposes an innovative smart contract language	First sharding protocol with presence of Byzantine fault	BLS ^a^ consensus	State sharding (i.e., Harmony shards Blockchain state)	The miner of the next block is predictable
Drawbacks	High computational power and low performance	transactions sharding ^b^	Uses small committee size	Centralization concern due to coinage	Centralization concern due to coinage	Centralization concern due to coinage
Advantages	High level of security	Uses PoW as identity registration to prevent Sybil attacks	Ensures good randomness	High level of privacy	Provides consistent cross-shard transactions	Agile architecture

✓: has property; ✕: does not have property; N/A: Not Available in the literature. ^1^ 8 shards; ^2^ 64 shards. ^a^ beside PoS and pBFT, Incognito [13] use an additional consensus, BLS; Incognito implements BLS for multi-signature aggregation; ^b^ each node has to hold the entire Blockchain state to be able to process transactions.

Incognito [13] claims significantly higher performance compared to other privacy Blockchains (e.g., Nxt [19] and Zcash [20]). Nxt [19] can usually handle less than 10 tx/s while Zcash handles only 6 tx/s.

## 3. Probabilistic Model

In this section, we propose a probabilistic model to analyze the security of a sharding-based PoS Blockchain protocols, called Incognito [13].

### 3.1. Notations & Architecture

This section describes the notations and definitions used to represent the proposed probabilistic model as well as the architecture of Incognito [13].

#### 3.1.1. Notations & Definitions

Table 2 shows the notations and symbols that are used throughout the paper.

**Definition** **1** (Faulty block)**.**
*A faulty block is a block that contains fraudulent transactions.*


**Definition** **2** (Conquering the Protocol)**.**
*A protocol is said to be conquered if the malicious nodes succeed in adding a faulty block to the blockchain.*


**Definition** **3** (Committee Resiliency of a Shard)**.**
*The maximum percentage of malicious nodes that the committee of the shard chain can contain while remaining secure.*


**Definition** **4** (Committee Resiliency of the Beacon Chain)**.**
*The maximum percentage of malicious nodes that the committee of the beacon chain can support while remaining secure.*


#### 3.1.2. Architecture of Incognito

Figure 1 shows the structure/architecture of Incognito [13], a sharding-based PoS Blockchain protocol. The network contains a single beacon chain and ζ shard chains. Each user/node can stake to be a validator either for the beacon chain or for the shard chain (see Figure 1). Shard chains produce blocks in parallel. All shard chains are synchronized by the beacon chain. More specifically, each shard has its own committee, which is randomly assigned by the beacon chain. Each shard chain processes a subset of the transactions submitted to the network. When a shard block is created, the beacon committee verifies the block; if it is valid, it adds the block header to the beacon chain. Otherwise, it drops it and sends the proof to other shards for a vote to slash the misbehaving shard committee. Furthermore, in each epoch, the beacon chain shuffles the committees of the shards to ensure security. For Incognito [13], when a new random number is generated, the beacon chain shuffles the committees; one epoch, for Incognito, corresponds to generating a new random number. This number is generated periodically in a round-robin fashion [13,21].

### 3.2. Probability Distributions

The main idea behind the sharding solution is to split/divide the network into subsets, called shards. Every single shard processes a subset of transactions rather than the entire network processing all transactions. This idea allows the network to scale (in terms of the number of transactions per second) with the number of shards. However, this technique may compromise the security of the network [8].

In Incognito [13], to add a faulty block to the Blockchain, it must be confirmed by at least ⌊βn⌋ (0≤β≤1; β=r) of the shard committee members, by at least $\left\lfloor \gamma n^{\prime }\right\rfloor$ of the beacon committee members ($\gamma =r^{\prime }$), and by at least ⌊δζ⌋ (0≤δ≤1) of all shards’ committees. For *Incognito* [13], $\beta =\delta =\frac{2}{3}$ and $\gamma =\frac{1}{2}$.

Let *X* be a random variable that represents the number of malicious nodes sampled in the committee of a shard from the validators in that shard. Each of the sampled nodes can be placed in one of 2 distinctive and disjoint groups; honest nodes group or malicious nodes group. Because the committees do not overlap, we have sampling without replacement. Thus, we conclude that *X* follows the hypergeometric distribution with parameters V, *M*, and *n* (it can be written as follows: X∼H(V,M,n)).

The probability mass function corresponds to *X* can be defined as follows [22]:(1)P(X=ω)=MωV−Mn−ωVn
where max(0,n−V)≤ω≤min(M,n) and V−M=H.

**Lemma** **1.**
*The probability of a shard’s committee to commit a faulty block (P) can be expressed as follows:*

(2)
P(X≥⌊βn⌋)=∑i=⌊βn⌋nMiV−Mn−iVn



Proof of Lemma 1 results directly from the cumulative hypergeometric distribution [8,11].

Similarly to *X*, $X^{\prime }$ follows the hypergeometric distribution with parameters $V^{\prime }$, $M^{\prime }$, and $n^{\prime }$.

**Lemma** **2.**
*The probability of at least ⌊δζ⌋ shards committees committing a faulty block ($P^{\prime }$) can be computed as follows:*

(3)
∑k=⌊δζ⌋ζ(P(X≥⌊βn⌋))k=∑k=⌊δζ⌋ζ∑i=⌊βn⌋nMiV−Mn−iVnk



**Proof of Lemma** **2.**The minimum number of committees to commit a faulty block is ⌊δζ⌋, where ζ is the number of shards. The probability of exactly ⌊δζ⌋ committees confirm/agree to add a faulty block can be expressed as follows:
(4)$$P_{\left\lfloor \delta \zeta \right\rfloor }={(P(X\geq \lfloor \beta n\rfloor ))}^{\left\lfloor \delta \zeta \right\rfloor }$$The probability to commit a faulty block by exactly ⌊δζ⌋+1 committees can be expressed as follows:
(5)$$P_{\left\lfloor \delta \zeta \right\rfloor }+1={(P(X\geq \lfloor \beta n\rfloor ))}^{\lfloor \delta \zeta \rfloor +1}$$Similarly, the probability of exactly ζ committees (the entire number of shards in this case) agreeing to add a faulty block can be expressed as follows:
(6)$$P_{\zeta }={(P(X\geq \lfloor \beta n\rfloor ))}^{\zeta }$$
A faulty block can be committed if ⌊δζ⌋ or ⌊δζ⌋+1 or ⌊δζ⌋+2, ⋯, or ζ committees agree to add this block. This can be mathematically computed by the sum over all these probabilities and can be expressed as follows:
(7)$$\begin{array}{cc} P^{\prime \prime } & =P_{\left\lfloor \delta \zeta \right\rfloor }+P_{\lfloor \delta \zeta \rfloor +1}+\cdots +P_{\zeta } \end{array}$$
(8)$$\begin{array}{cc} \; & =\sum_{i=0}^{\zeta -\lfloor \delta \zeta \rfloor }P_{\lfloor \delta \zeta \rfloor +i} \end{array}$$□

**Lemma** **3.**
*The probability of the beacon’s committee committing a faulty block ($P^{\prime \prime }$) can be expressed as follows:*

(9)
P(X′≥⌊γn′⌋)=∑j=⌊γn′⌋n′M′jV′−M′n′−jV′n′

*where $V^{\prime }-M^{\prime }=H^{\prime }$.*


Proof of Lemma 3 results directly from the cumulative hypergeometric distribution [8,11].

**Theorem** **1** (Committing a Faulty Block)**.**
*The probability of committing a faulty block ($P_{f}$) by a given sharding-based PoS Blockchain protocol can be expressed as follows:*

(10)
Pf=∑k=⌊βn⌋n∑j=⌊γn′⌋n′∑α=⌊δζ⌋ζMkHn−kM′jH′n′−jVnV′n′∑i=⌊βn⌋nMiV−Mn−iVnα



**Proof of Theorem** **1.**To commit a faulty block, it must be confirmed/verified by at least ⌊βn⌋ of the shard committee members, by at least $\left\lfloor \gamma n^{\prime }\right\rfloor$ of the beacon committee members, and by at least ⌊δζ⌋ of all shards’ committees. This can be expressed by the product over the three probabilities (the calculated probabilities in Lemmas 1, 2, and 3). □

### 3.3. Years to Fail

To make the measurement of the security more readable, we propose to compute the average number of years to fail ($Y_{f}$) based on the calculated failure probability (i.e., the probability of conquering the protocol). This number can be expressed as follows:(11)$$Y_{f}=\frac{1}{P_{f}N_{s}}$$
where $P_{f}$ is the probability of committing (adding) a faulty block to the blockchain and $N_{s}$ is the number of sharding rounds per year (also referred to as the number of epochs per year).

## 4. Evaluation Results

In this section, we evaluate the effectiveness of the proposed probabilistic model via numerical simulations.

### 4.1. Simulation Setup

In order to implement the proposed probabilistic model, we make use of a built-in Python library called **SciPy**. We import **hypergeom**, a hypergeometric discrete random variable, from the **scipy.stats** module. Specifically, we use the **cdf** (i.e., cumulative distribution function) function. We also make use of the **math** module, which provides access to mathematical functions (e.g., the **floor** function). For the results plot, we make use of **matplotlib.pyplot** library.

### 4.2. Results and Analysis

In Figure 2, we assume a network with N= 2000 nodes, V = 200, $V^{\prime }$ = 400, ζ = 8, *r* = $r^{\prime }$ = 0.5. We chose these values for *r* and $r^{\prime }$ to make the results more readable. However, we can use other values (e.g., Incognito [13] values). Figure 2 illustrates how the committee size of the shard impacts its failure probability P as well as the failure probability of all shards $P^{\prime }$ and the relationship between the committee size of the beacon chain and its failure probability $P^{\prime \prime }$. In particular, Figure 2a shows the probability of a shard to commit a faulty block versus the size of the committee. We observe that the probability P decreases when the size of the committee increases. More specifically, we observe that the probability corresponding to R = 0.2 (i.e., 20% of malicious nodes in each shard) decreases rapidly compared to those of R = 0.25 and R = 0.3; this can be explained by the small percentage of malicious nodes. In other words as the percentage of malicious nodes decreases, so does the probability.

Figure 2b shows the probability of all shards committing a faulty block versus the size of the committee. We observe that the probability $P^{\prime }$ decreases when the size of the committee increases. Similarly, as the percentage of malicious nodes slightly increases in the shard, the probability of committing a faulty block increases.

Figure 2c shows the probability of the beacon chain to commit a faulty block ($P^{\prime \prime }$) versus the size committee of the beacon chain ($n^{\prime }$). We also observe that the probability $P^{\prime \prime }$ decreases when the size of the committee increases. More specifically, we observe that the probability corresponding to R=0.2 (i.e., 20% of malicious nodes in the beacon chain) decreases sharply compared to those of R=0.25 and R=0.3.

In Figure 3, we assume a network with 2000 nodes, V=200, $V^{\prime }=400$, ζ = 8, $n^{\prime }=100$. Figure 3a shows the probability of conquering the protocol when varying the committee size of the shard. We observe that as the committee size of the shard increases the probability of conquering the protocol decreases. Figure 3b shows the number of years to fail ($Y_{f}$) versus the committee size of the shard. We observe that when the committee size of the shard increases, the number of years to fail increases as well.

Table 3 shows the probability of conquering the chain (i.e., the probability of committing a faulty block; it is calculated based on Theorem 1) for different percentages of malicious nodes in the shards as well as in the beacon chain. Moreover, Table 3 shows the number of years to fail corresponding to these probabilities. We observe that as the percentage of malicious nodes increases the number of years to fail decreases. More specifically, we observe that the probability of conquering the chain is extremely low even with 20% of the malicious nodes in each shard as well as in the beacon chain. This achieves good security, reaching about 1.74E+17 years to fail.

Table 4 shows the trade-off between the values of *n* and $n^{\prime }$ that provide certain target value of years to fail. We consider a network with 2000 nodes, V=200, $V^{\prime }=400$, ζ = 8, $R=R^{\prime }=0.3$, and $\delta =r=r^{\prime }=0.33$. To reach a level of security corresponding to 2500 years to fail, you should set *n* to 95 and $n^{\prime }$ to 150 or *n* to 85 and $n^{\prime }$ to 145. However, you should adjust the values of *n* and $n^{\prime }$ carefully to reach the desired target. Similarly, if you have a network with 4000 nodes, V=300, $V^{\prime }=1000$, ζ = 10, $R=R^{\prime }=0.3$, and $\delta =r=r^{\prime }=0.33$ and set years to fail to as 4000, we can achieve it by setting *n* to 100 and $n^{\prime }$ to 250 or *n* to 122 and $n^{\prime }$ to 130. It is evident that there are different possibilities and combinations of the values of *n* and $n^{\prime }$. However, you should adjust these values (*n* and $n^{\prime }$) carefully to reach the desired target (the desired number of years to fail ($Y_{f}$)).

Finally, we conclude that by adjusting the committee size of the shard as well as the committee size of the beacon chain, we can protect sharded Blockchain systems (based on PoS) against malicious nodes (e.g., Sybil nodes).

## 5. Limitations of the Paper and Future Work

Our study has some limitations, including the fact that the proposed probabilistic model is similar to that of Incognito [13]. However, each sharding-based PoS Blockchain protocol bears a specific structure, which is slightly different from other sharding-based PoS Blockchain protocols (e.g., Harmony [17]). Furthermore, this study does not provide an in-depth literature review of sharding-based PoS Blockchain protocols since the objective was to provide a brief and simple presentation of the key components of these protocols. The aim of doing so was to prepare the readers to understand how we can deal with the security analysis of these protocols using probability distributions.

Future works will focus on analyzing the security of novel types of sharding-based Blockchain protocols including DankSharding by Ethereum [23], Shardus [24] and probabilistic analysis of cross-shard transactions.

## 6. Conclusions

In this paper, we first illustrate the key components of sharding-based PoS Blockchain protocols. Next, we discuss two consensus mechanisms, PoS and pBFT. We also address the security of Incognito, a sharding-based PoS Blockchain protocol. In particular, we provide a probabilistic model to compute the probability of committing a faulty block. Based on this probability, we compute the number of years to fail. Furthermore, this article depicts that we can control the number of years to fail by adjusting the size of the shard as well as the size of the beacon committee. Our future work includes the computation of the failure probability across shard transactions.

## Figures and Tables

**Figure 1 sensors-23-02819-f001:**
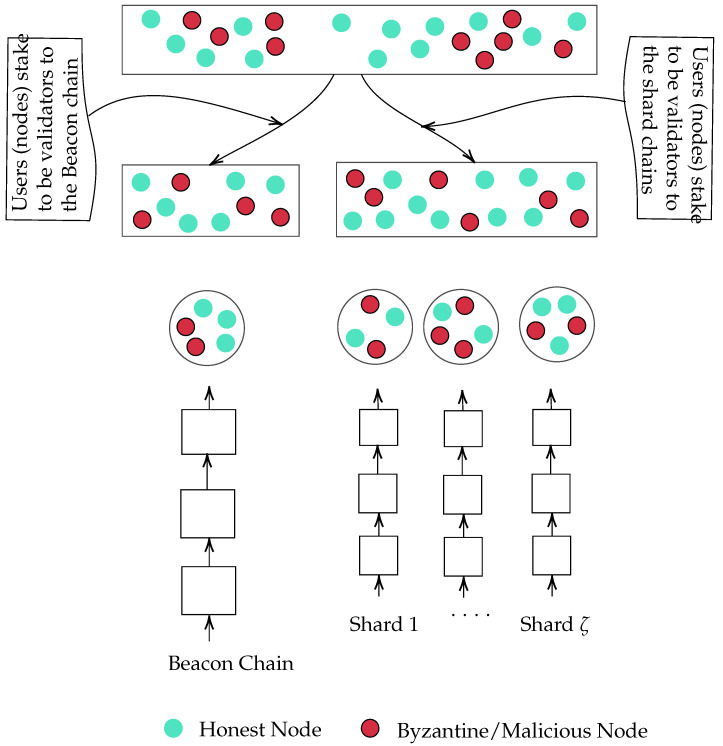
A sharding-based PoS and pBFT Blockchain protocol.

**Figure 2 sensors-23-02819-f002:**
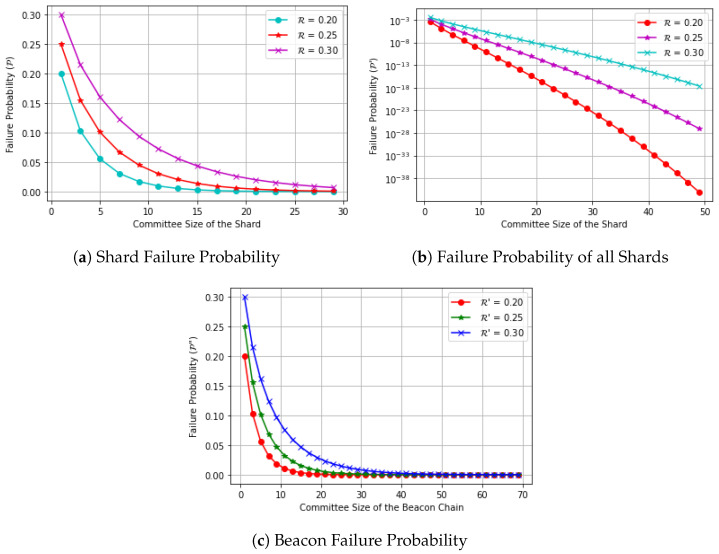
(**a**) Probability of a shard to commit a faulty block (P) versus the committee size of the shard (*n*), (**b**) Log-scale plot of the probability of all shards committing a faulty block ($P^{\prime }$) versus the size of the committee (*n*), and (**c**) Probability of the beacon chain to commit a faulty block ($P^{\prime \prime }$) versus the size committee of the beacon chain ($n^{\prime }$).

**Figure 3 sensors-23-02819-f003:**
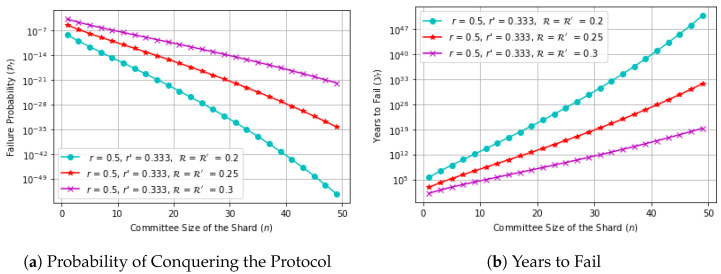
Log-scale plot: (**a**) Probability of conquering the protocol ($P_{f}$) versus the committee size of the shard (*n*), (**b**) Number of years to fail ($Y_{f}$) versus the committee size of the shard (*n*).

**Table 2 sensors-23-02819-t002:** Notations & Symbols.

Notation	Description
N	Number of users (aka number of nodes)
*n*	Committee size of a shard
$n^{\prime }$	Committee size of the beacon chain
*H*	Number of honest validators in a shard
*M*	Number of malicious validators in a shard
V	Number of validators in a shard (V=H+M)
ζ	Number of shards
*X*	Random variable that computes the number of malicious nodes in the committee of a shard
$H^{\prime }$	Number of honest validators in the beacon chain
$M^{\prime }$	Number of malicious validators in the beacon chain
$V^{\prime }$	Number of validators in the beacon chain ($V^{\prime }=H^{\prime }+M^{\prime }$)
$X^{\prime }$	Random variable that computes the number of malicious nodes in the committee of the beacon chain
*r*	Resiliency of the shard committee, 0≤r≤1
$r^{\prime }$	Resiliency of the beacon committee, $0\leq r^{\prime }\leq 1$
*R*	Percentage of malicious validators in a shard chain
$R^{\prime }$	Percentage of malicious validators in the beacon chain
$P_{f}$	Probability of conquering the protocol
P	Probability of a shard committing a faulty block
$P^{\prime }$	Probability of all shards committing a faulty block
$P^{\prime \prime }$	Probability of the beacon chain committing a faulty block
$Y_{f}$	Number of years to fail

**Table 3 sensors-23-02819-t003:** Probability of conquering the protocol.

R = $R^{\prime }$	10%	15%	20%	30%
$P_{f}$ ^1^	3.63E-66	2.10E-34	1.58E-18	1.70E-04
$Y_{f}$ ^1^	7.56E+62	1.30E+31	1.74E+17	16.12
$P_{f}$ ^2^	0.0	5.14E-80	2.01E-41	5.30E-07
$Y_{f}$ ^2^	inf	5.33E+76	1.36E+38	5171.32

^1^ Scenario 1. ^2^ Scenario 2.

**Table 4 sensors-23-02819-t004:** Potential realistic scenarios.

N	V	$V^{\prime }$	ζ	R	$R^{\prime }$	δ	*r*	$r^{\prime }$	*n*	$n^{\prime }$	Target ($Y_{f}$)
2000	200	400	8	0.3 (30%)	0.3 (30%)	0.33 (33%)	0.33 (33%)	0.33 (33%)	95	150	≈2500
2000	200	400	8	0.3 (30%)	0.3 (30%)	0.33 (33%)	0.33 (33%)	0.33 (33%)	85	145	≈2500
4000	300	1000	10	0.3 (30%)	0.3 (30%)	0.33 (33%)	0.33 (33%)	0.33 (33%)	100	250	≈4000
4000	300	1000	10	0.3 (30%)	0.3 (30%)	0.33 (33%)	0.33 (33%)	0.33 (33%)	122	130	≈4000

## Data Availability

Not applicable.

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
