# Peer review of "Sharding-Based Proof-of-Stake Blockchain Protocols: Key Components & Probabilistic Security Analysis†"

_sensors, 2023, doi:10.3390/s23052819_

Round 1

Reviewer 1 Report

Title: Sharding-Based Proof-of-Stake Blockchain Protocols: Key Components & Probabilistic Security Analysis

1-      A novel probabilistic model was proposed to analyse the security of protocols described in the paper. The study begins by introducing the key components of PoS blockchain protocols based on sharding. The section then provides a brief introduction to two consensus mechanisms, namely Proof-of-Stake (PoS) and practical Byzantine Fault Tolerance (pBFT), and discusses their application and limitations in the context of sharding-based blockchain protocols. The research paper then presents a probabilistic model for evaluating the security of these protocols. Specifically, they compute the probability of committing a faulty block and measure the security by calculating the number of years it will take for the block to fail. The study concludes with a numerical evaluation of the effectiveness of the proposed model. Hence, I agree that the proposed model provides a novel solution and contributes to the body of knowledge.

2-      The abstract of a research paper should also include a statement of the problem, the purpose of the study, an analysis of the data, the results, and a conclusion. In this section, minor modifications are suggested.

3-      The study's purpose is inconsistently stated throughout the paper. The research must have a distinct focus and objective, which must be reflected throughout the entire manuscript. Therefore, it is strongly suggested that Section 2.2's architecture presentation be improved.

4-      You have provided insufficient details regarding the design and architecture employed. Did you evaluate the design's dependability and validity? Before the method section, please state the research questions that guided the study. I believe that the methods section needs to be reorganised. Start with the sample's description. Then, discuss the recruitment of the sample. Then discuss your research methods.

5-      The simulation results in Section 4.1 were extremely brief and did not describe the method selected to conduct the performance and evaluation tests.

6-      You need to include a section on limitations of the study and ideas for future research.

Author Response

Dear Reviewer,

Thanks,

Abdelatif

Reviewer 2 Report

The domain and the title of the research is quite good and it provides important information to the users, but still the following issues should be revised. Minor revision is required with the following comments.

I accept this paper with minor revisions.

1.       The paper needs thorough proofreading and some typos should be removed.

2.       The author adds some figure that highlight the working of blockchain as well as smartcontracts so that the general can understand the use and application of blockchain technology in healthcare.

3.       Figures need revision to be more high quality.

4.       The following reference should be added which provides a deeper insight on blockchain-based healthcare system.

Almaiah, M. A., Hajjej, F., Ali, A., Pasha, M. F., & Almomani, O. (2022). A Novel Hybrid Trustworthy Decentralized Authentication and Data Preservation Model for Digital Healthcare IoT Based CPS. Sensors, 22(4), 1448. https://www.mdpi.com/1424-8220/22/4/1448

Author Response

Dear Reviewer,

Thanks,

Abdelatif

Reviewer 3 Report

The authors analyse a sharding-based proof-of-stake blockchain model with two consensus mechanisms and provide a mathematical analysis of the security of these protocols.

The work is timely and interesting, but it has a major problem. It's almost entirely the same as the authors' previous work.

- Abstract almost completely the same
- Almost all of the introduction section
- Section 2 is new (but its content was in different places on the previous paper)
- Section 3 also almost the same with the exception of the "Years to Fail" subsection
- Section 4 also almost all the same with the exception of table 4 and the accompanying description
- Even the conclusion is mostly the same

The novel part seems to be the "Years to Fail" computation and without some other form of added content does not seem enough to be considered a new paper.

If this work could be contextualized differently with attention to not keeping exact phrases from the previous work and focusing only on the new achievements it might salvageable but at this time I can't recommend publishing it.

Author Response

Dear Reviewer,

Thanks,

Abdelatif

Round 2

Reviewer 1 Report

The new manuscript also address the reviewers’ concerns and comments. Therefore, I concur that the suggested model offers a fresh approach and adds to the body of knowledge. The revised manuscript also addresses the issues and criticisms raised by the reviewers.

Reviewer 3 Report

Thank you to the authors for this reply.

As I've stated to the authors previously, using verbatim phrases from
other published works, even in a paper intended as a conference
extension, seems like low effort.

Still the authors added in this version a few novel findings, so my
recommendation is to accept the manuscript as is.